# Cloning, Expression and Inhibitory Effects on Lewis Lung Carcinoma Cells of rAj-Tspin from Sea Cucumber (*Apostichopus japonicus*)

**DOI:** 10.3390/molecules27010229

**Published:** 2021-12-30

**Authors:** Rong Qiao, Rong Xiao, Zhong Chen, Jingwei Jiang, Chenghua Yuan, Shuxiang Ning, Jihong Wang, Zunchun Zhou

**Affiliations:** 1Liaoning Key Lab of Marine Fishery Molecular Biology, Liaoning Ocean and Fisheries Science Research Institute, Dalian 116023, China; qrrrr95@163.com (R.Q.); ch_zhong@163.com (Z.C.); weijingjiang1984@163.com (J.J.); 2School of Life Sciences, Liaoning Normal University, Dalian 116081, China; xiaorong@lnnu.edu.cn (R.X.); yuanchenghua513@163.com (C.Y.); NSX@lnnu.edu.cn (S.N.)

**Keywords:** sea cucumber, ADAMTS13-like, TSP 1, RGD, anti-tumoral activity, rAj-Tspin, LLC cells

## Abstract

In recent years, sea cucumber has become a favorite healthcare food due to its characteristic prevention of cardiovascular diseases, suppression of tumors, as well as enhancement of immunity. In order to screen the anti-tumoral proteins or peptides from sea cucumber (*Apostichopus japonicus*), its cDNA library was analyzed, and a disintegrin-like and metalloproteinase with thrombospondin type 1 motif, member 13 (ADAMTS13)-like was found. ADAMTS13-like contains 10 thrombospondin 1 (TSP1) domains. Based on analysis of bioinformatics, the third TSP1 domain of this protein, which is further named Aj-Tspin, contains an arginine-glycine-aspartate (RGD) motif. Since our previous studies showed that the recombinant RGD-containing peptide from lampreys showed anti-tumoral activity, the third TSP1 domain of ADAMTS13-like was chosen to evaluate it’s effect on tumor proliferation and metastasis, despite the fact it shares almost no homologue with disintegrins from other species. After artificial synthesis, its cDNA sequence, Aj-Tspin, which is composed of 56 amino acids, was subcloned into a pET23b vector and expressed as a recombinant Aj-Tspin (rAj-Tspin) in a soluble form with a molecular weight of 6.976 kDa. Through affinity chromatography, rAj-Tspin was purified as a single protein. Both anti-proliferation and immunofluorescence assays showed that rAj-Tspin suppressed the proliferation of Lewis lung carcinoma (LLC) cells through apoptosis. Adhesion assay also displayed that rAj-Tspin inhibited the adhesion of LLC cells to ECM proteins, including fibronectin, laminin, vitronectin and collagen. Lastly, rAj-Tspin also suppressed the migration and invasion of LLC cells across the filter in transwells. Thus, the above indicates that rAj-Tspin might act as a potential anti-tumoral drug in the future and could also provide information on the nutritional value of sea cucumber.

## 1. Introduction

A disintegrin-like and metalloproteinase with thrombospondin type 1 motif, member 13 (ADAMTS13) belongs to the ADAMTS family, which is composed of 19 proteases [1]. Except the domains shared by the 19 ADAMTS members, human ADAMTS13 has eight thrombospondin type 1 motifs and has been most reported to affect the progress of thrombotic thrombocytopenic purpura [1,2]. Thrombospondin-1 (TSP1), also called thrombin-sensitive protein, is a glycoprotein that belongs to the extracellular matrix (ECM) protein family [3,4]. Previous studies have shown that TSP1 is expressed in a variety of tissues and is composed of six domains, which could lead to the interaction with multiple cell-surface molecules, including CD36, integrins, etc. [4,5,6]. Thus, it is not surprising that TSP1 was not only reported to participate in regulation of the process of angiogenesis by affecting proliferation, migration and invasion behavior of endothelial cells but also to control the process of tumor occurrence and development by affecting proliferation, mobility, apoptosis and immunity of tumor cells [4].

As one of the most common causes of death worldwide, lung cancer accounts for approximately 1.6 million deaths annually [7]. Approximately 85% of these patients suffer from the histological subtype of non-small-cell lung cancer (NSCLC) [8]. Unfortunately, data from most countries indicates that the incidence of this type of cancer is increasing continuously year by year [9]. During the past 20 years, identification of the biological characteristics of NSCLC and its treatment strategies have progressed to a significant degree. Both targeted therapy and immunotherapy have provided novel information for treatment with controllable toxicity, which could prolong the lifespan of NSCLC patients. However, cure rates of NSCLC patients, especially those with metastatic tumors, remain low. Therefore, more effective drugs targeted for NSCLC patients are still required to be screened and verified.

Sea cucumber (*Apostichopus japonicus*, *A. japonicus*), which belongs to the phylum Echinodermata, is an important kind of economic aquaculture species in Liaoning Province [10]. In recent years, sea cucumber has become a favorite healthcare food due to its characteristic prevention of cardiovascular diseases, suppression of tumors, as well as enhancement of immunity [11,12,13]. In order to clarify the anti-tumoral activity and mechanisms of sea cucumber, its cDNA library was analyzed, and an ADAMTS13-like was found in relative abundance. According to the analysis of domain by the EXPASY website, ADAMTS13-like was found to contain 10 TSP1 domains. Among the 10 TSP1 domains, the third TSP1 domain contains an arginine-glycine-aspartate (RGD) motif, which is further named Aj-Tspin in the present study. As TSP1 plays contrary roles during tumor progression, whether Aj-Tspin exhibits anti-tumoral activity aroused our interest. Therefore, the cDNA of Aj-Tspin was artificially synthesized, cloned and expressed in the prokaryotic expression system. Additionally, the effect of its anti-tumoral activity on mouse Lewis lung cancer (LLC) cells was clarified.

## 2. Results

### 2.1. A ADAMTS13-Like Was Identified from A. japonicus

During the analysis of the cDNA library from sea cucumber (*A. japonicus*), several cDNA sequences showing homologues with ADAMTS13 were identified. After splicing the cDNA sequences and finding the open reading frame (ORF), the identified ADAMTS13-like from *A. japonicus* was composed of 2799 bp, which encodes 933 amino acids (Figure A1). Based on analysis of the phylogenetic tree, the ADAMTS13-like from *A. japonicus* showed relatively significant relationship with the putative ADAMTS9 isoform X2 from *A. japonicus*, ADAMTS13-like isoforms X1, X2 and X3 from *Acanthaster planci*, ADAMTS15-like from *Patiria miniate*, as well as ADAMTS16-like from *Asterias rubens* (Figure 1).

### 2.2. Domain Analysis of ADAMTS13-Like from A. japonicus

According to sequence analysis through the EXPASY website, ADAMTS13-like from *A. japonicus* is composed of 10 TSP1 domains (Figure 2). Among the 10 TSP1 domains, only the third TSP1 domain was found to contain an RGD motif, which is the classic characterization of disintegrins (Figure A2). As our previous studies found that the recombinant RGD-containing protein from *Lampetra japonica* showes anti-tumoral activity obviously, we focused on the third domain of ADAMTS13-like from *A. japonicus* in order to determine whether this TSP1 domain would exhibit anti-tumoral characterization. Thus, the third TSP1 domain, which was named Aj-Tspin, was selected for further molecular cloning and functional studies. As shown in Figure A2, the nucleic-acid sequence of Aj-Tspin is composed of 168 bp, which encodes 56 amino acids. Based on domain prediction, the third TSP1 domain (Aj-Tspin) was characterized by an RGD motif and three pairs of disulfides (Figure A2). Furthermore, the theoretical molecular weight and the theoretical isoelectric point of rAj-Tspin with the His-tag were 6.976 kDa and 6.3, respectively.

### 2.3. Aj-Tspin Shares Low Homology with Disintegrins from Other Species

To date, disintegrins have mainly been found in five different species, including snakes [14,15,16,17], leeches [18,19], ticks [20,21], horseflies [22,23] and lampreys [24,25]. Based on acid sequence analysis, different disintegrins from snake venom show more than 60% sequence identities, while disintegrins from the five different species almost show no homology, except that the common characteristics are the RGD motif and several pairs of cysteines (Figure 3). Similarly, Aj-Tspin shows almost no homologue with the disintegrins from the above five species, as previously reported (Figure 3). Additionally, Aj-Tspin contains the RGD motif and several cysteines shared by disintegrin family members.

### 2.4. Aj-Tspin Was Synthesized and Expressed as a Soluble Protein

In order to further study its anti-tumoral activity, the cDNA sequence of Aj-Tspin was artificially synthesized and verified through sequencing. After subcloning into a pET23b vector and inducing with isopropyl-1-thio-β-D-galactopyranoside (IPTG), the recombinant Aj-Tspin (rAj-Tspin) was mainly expressed as a soluble protein with a His-tag (Figure 4). When rAj-Tspin was purified through affinity chromatography, a single protein band on the Tricine sodium dodecyl sulfate-polyacrylamide gel electrophoresis (SDS-PAGE) was observed (Figure 4), and the purity of the purified rAj-Tspin was approximately 95%. In addition, Coomassie Brilliant Blue (G250) assay showed the concentration of the purified rAj-Tspin was approximately 0.59 mg/mL.

### 2.5. rAj-Tspin Dose-Dependently Suppressed Proliferation of LLC Cells 

Here, the proliferation of LLC cells in the presence of rAj-Tspin was detected by CCK-8 assay. As shown in Figure 5, proliferation of LLC cells was inhibited by rAj-Tspin. As the dosage of rAj-Tspin (0.37, 0.75, 1.12, 1.49, 1.86, 2.24, 2.98, 3.35 and 3.73 μM) increased, the proliferation of LLC cells was suppressed evidently, indicating that the inhibition induced by rAj-Tspin was dose-dependent. Based on the calculation from the formula, the half maximal inhibitory concentration (IC_50_) of rAj-Tspin on the proliferation of LLC cells was 1.85 µM.

### 2.6. rAj-Tspin Affected the Shape of LLC Cells and Induced Apoptosis

According to the IC_50_ from CCK-8 assay, 1.85, 2.20 and 3.04 μM rAj-Tspin were chosen to further treat the LLC cells. As shown by Wright-Giemsa staining assay, the number and density of LLC cells decreased evidently under the microscope when the concentration of rAj-Tspin increased (Figure 6). In the control groups, the LLC cells mainly extended in two forms: short, spindle shape and irregular, round shape. LLC cells were likely to aggregate and form dense clones (Figure 6). After treatment with rAj-Tspin, the shape of LLC cells became round and the connection between the LLC cells gradually disappeared. In the presence of 3.04 μM rAj-Tspin, LLC cells were not liable to aggregate (Figure 6). As shown in Figure 7, terminal deoxynucleoitidyl transferase-mediated dUTP nick end labeling (TUNEL) assay showed that the fluorescent intensity in the empty-medium-treated LLC cells was very weak and was almost undetectable, indicating that the LLC cells were in their normal state (Figure 7). However, the fluorescent intensity in the LLC cells became stronger as the concentration of rAj-Tspin increased, indicating that the number of apoptotic LLC cells increased (Figure 7). Thus, rAj-Tspin induced the apoptosis of LLC cells in a dose-dependent manner. As shown in Figure 8, the LLC cells treated with empty-medium extended well with short spindles or were in the round state. At this time, the cytoskeleton and nuclei of the LLC cells were intact, which could be observed clearly. As the dosage of rAj-Tspin increased, microfilaments in the LLC cells were gradually destroyed until the cytoskeleton was completely disintegrated in the 3.04 μM rAj-Tspin groups (Figure 8). Meanwhile, the nuclei in the LLC cells treated with rAj-Tspin shrank and exhibited strong blue fluorescence, indicating that the chromatin in the LLC cells was condensed, which is a typically apoptotic phenomenon (Figure 8). Therefore, the above results suggest that rAj-Tspin destroyed the cytoskeleton of LLC cells and induced their apoptosis.

### 2.7. Inhibitory Effects of rAj-Tspin on Adhesion of LLC Cells to ECM Proteins

Here, interaction of LLC cells with ECM proteins was detected through CCK-8 assay. As shown in Figure 9, the ability of LLC cells to adhere to the four ECM proteins, including fibronectin (FN), laminin (LN), vitronectin (VN) and collagen (Col) constantly reduced as the dosages of rAj-Tspin increased. When 3.04 µM rAj-Tspin was added in the same 96-well plates, the ability of the LLC cells to adhere to the above four ECM proteins was lowest among the three dosages, which further indicates that the inhibition of adhesion of LLC cells to the four kinds of ECM proteins induced by rAj-Tspin is dose dependent. The adhesive rate of LLC cells in the presence of 1.85, 2.20 and 3.04 µM rAj-Tspin is shown in Table 1.

### 2.8. Inhibitory Effects of rAj-Tspin on Mobility of LLC Cells

According to the previous studies, transwell assays were also used to detect the inhibitory effects of rAj-Tspin on the migration and invasion ability of LLC cells [24]. As shown in Figure 10, the number of migrated LLC cells in the empty-medium groups that were not treated with the inducer basic fibroblast growth factor (bFGF) was smaller than that in the empty-medium groups treated with bFGF. In addition, the average number of migrated LLC cells treated with 0, 1.85, 2.20 and 3.04 µM rAj-Tspin in the presence of bFGF was 191, 124, 117 and 69, respectively (Figure 10). Furthermore, the migration rate of LLC cells treated with both empty-medium and bFGF was set as 100%. Based on the formula, the inhibitory migration rate of LLC cells in the presence of 1.85, 2.20 and 3.04 µM was 35.25%, 38.92%, and 63.87%, respectively. Similarly, the number of invaded LLC cells in the empty-medium groups that were not treated with the inducer bFGF was smaller than that in the empty-medium groups treated with bFGF. Furthermore, the average number of invaded LLC cells treated with 0, 1.85, 2.20 and 3.04 µM rAj-Tspin in the presence of bFGF was 64, 29, 26 and 7, respectively (Figure 11). Moreover, the invasion rate of LLC cells treated with both empty-medium and bFGF was set as 100%. Finally, the inhibitory invasion rate of LLC cells in the presence of 1.85, 2.20 and 3.04 µM rAj-Tspin was 54.17%, 59.38% and 89.58%, respectively.

## 3. Discussion

Sea cucumbers are healthcare products that, in recent years, have been considered to possess anti-tumor activity and the ability to enhance immunity [11,26,27]. Especially in the Asia and Middle East, sea cucumbers are taken as food or folk medicine by residents to prevent cancer and to enhance immunity [28]. However, the recombinant peptides from *A. japonicus* that account for the anti-tumor activity of sea cucumbers, as well as the detailed mechanisms have not yet been reported. According to previous studies, the disintegrins from snakes and lampreys were reported to inhibit tumor progression, as they contain RGD motifs which could bind to integrins on the tumor-cell membrane [14,15,16,24,25]. Therefore, in the present study, disintegrins with RGD motifs were firstly screened from the cDNA library of sea cucumber, and an ADAMTS13-like was found, which contains 10 TSP1 domains and has one RGD motif in its third TSP1 domain. Although sequence analysis showed the third TSP1 domain of the ADAMTS13-like from sea cucumbers shares low identities with the disintegrins from the other species, it contains an RGD motif, which indicates that the third TSP1 domain of ADAMTS13-like might possess anti-tumor activity like that of disintegrins from other species [14,15,16,24,25]. More importantly, human TSP-1 protein was also reported to regulate tumor progression [4]. Thus, we chose the cDNA sequence of the third TSP1 domain of ADAMTS13-like (Aj-Tspin) with the RGD motif to artificially synthesize and successfully obtained the soluble rAj-Tspin with a single band on the Tricine SDS-PAGE. As we all know, the source of active substances in marine species has restricted their application, a problem which needs to be solved urgently. In the present study, preparation of rAj-Tspin through genetic engineering technology provided the possibility of rAj-Tspin application as a potential anti-tumor drug that is not only low-cost but also environmentally friendly.

ADAMTS13 belongs to the ADAMTS family, deficiency of which could lead to thrombotic thrombocytopenic purpura [1,2]. Additionally, abnormal levels of ADAMTS13 were found to participate in the progress of the thrombotic microangiopathies, stroke and cardiovascular diseases [29]. In 2019, Takaya and colleagues reported that an imbalance of ADAMTS13 with its substrate, von Willebrand factor, could be used as a detection marker for diagnosis of hepatocellular carcinoma [30]. When compared with healthy groups, levels of ADAMTS13 decreased in the sera/plasma of advanced NSCLC patients [31]. In the present study, the full cDNA sequence of ADAMTS13-like from *A. japonicus* was not chosen for synthesis, as it contains 2799 bp, which encodes 933 amino acids with a predicted molecular weight of 102.5 kDa. At present, it is very difficult to obtain the recombinant protein in a soluble form with such a large molecular weight. More importantly, the recombinant protein with relatively larger molecular weight might cause serious problems, such as immunogenicity, during its application in clinical tests [32]. Thus, we chose only the third TSP1 domain of ADAMTS13-like from *A. japonicus* for synthesis, as it contains an RGD motif and is only 168 bp long which encodes 56 amino acids, with a molecular weight of 6.976 kDa. Furthermore, the possibility of immunogenicity induced by rAj-Tspin might decrease with a relatively lower molecular weight.

In previous studies, modified thymosin α 1, which contains an RGD motif, was reported to inhibit the growth of lung cancer in vivo [33]. Furthermore, a great number of studies have shown that TSP1 might exhibit opposite effects on tumor progression, probably due to its interaction with different receptors on the cells [34]. Compared with the tissues of healthy species, levels of TSP1 decreased, especially in late-stage NSCLC patients [35]. In 2016, Rouanne and colleagues found that increased levels of TSP1 reduced death risks in NSCLC patients, suggesting that TSP1 might act as an effective prognostic marker [36]. The above indicates that TSP1 might participate in the progression of NSCLC. Therefore, the question is raised whether rAj-Tspin, which possesses both the TSP1 domain and an RGD motif, could also inhibit the NSCLC progression. In our present study, rAj-Tspin was found to dose-dependently inhibit the proliferation, adhesion, migration and invasion of LLC cells, which are typical NSCLC cell line [37]. Similarly to the other disintegrins, rAj-Tspin could lead to apoptosis of LLC cells, as rAj-Tspin increased green fluorescence in LLC cells in TUNEL assay and induced chromatin concentrations, according to microscopic observation. Thus, this suggested that rAj-Tspin is a valid anti-tumor recombinant peptide that might be one of the main factors accounting for the anti-tumor effects of sea cucumber. 

Integrins are a type of protein, which promote adhesion of tumor cells to ECM proteins, including FN, LN, Col IV and VN [38,39,40,41]. Additionally, the RGD motifs in ECM proteins are able to interact with specific integrins on tumor cells, which ultimately cause adhesion of the tumor cells [42]. In the present study, rAj-Tspin was found to suppress adhesion of LLC cells to the above four ECM proteins, suggesting that rAj-Tspin could competitively bind to integrins, which ultimately interfered with the binding of tumor cells to ECM proteins. Furthermore, TSP1 was also reported to be associated with adhesion [4]. Thus, the adhesion of LLC cells to ECM proteins was suppressed, probably due to the RGD motif and TSP1 domain contained in rAj-Tspin.

Combining the data from the present study, rAj-Tspin effectively repressed the proliferation and mobility of LLC cells in in vitro experiments. As rAj-Tspin is a recombinant protein from sea cucumber with a relatively lower molecular weight, it could be obtained at low cost. However, many studies are still required to assess whether it could be used as a potential drug to treat NSCLC patients in the future, including the detailed signal pathway and the anti-tumor mechanism of rAj-Tspin on the LLC cells, the anti-tumoral activity of rAj-Tspin in vivo studies, as well as safety evaluation of rAj-Tspin to exclude immune problems, etc. 

## 4. Materials and Methods

### 4.1. Bioinformatics Analysis

Based on a previously constructed cDNA library from sea cucumber (*A. japonicus*), a homologue of ADAMTS13 was found through sequence assembly. In addition, both the NCBI database (https://www.ncbi.nlm.nih.gov/, accessed on 18 February 2020) and EXPASY (https://www.expasy.org/, accessed on 18 February 2020) were used to search the ORF, phylogenetic tree and domains of ADAMTS13-like from *A. japonicus*, respectively. Furthermore, multiple-sequence alignment of the third TSP1 domain of ADAMTS13-like was performed using Clustal O.

### 4.2. Cloning and Expression of Aj-Tspin

According to sequence analysis of ADAMTS13-like from sea cucumber (*A. japonicus*), the third TSP1 domain, which also contains an RGD motif, was synthesized by GenScript (Nanjing, China). Based on the above sequence, a pair of primers was designed that could be recognized by two restriction endonucleases, *Nde* I and *Hind* III, and listed as follows: 

5′-XXCATATGGCGCCGGCGCAGTGGGTGAAA-3′; 

5′-XXAAGCTTCCGACAATCCCCGTTGTTACA-3′. 

Subsequently, the PCR products of the third TSP1 domain of ADAMTS13-like from sea cucumber (*A. japonicus*), which was also named Aj-Tspin, were amplified through PCR, with the synthesized sequence as a template, and extracted through TaKaRa MiniBEST Plasmid Purification Kit Ver.4.0 (TaKaRa, Dalian, China) according to the manufacturer’s instructions. After ligating with pET23b, the recombinant plasmid (pET23b-Aj-Tspin) was transformed into *Escherichia coli* (*E. coli*) BL21 cells using the CaCl_2_ method [43]. Then, the positive bacteria with pET23b-Aj-Tspin were screened and identified through sequencing with T7 universal primers. Next, rAj-Tspin was expressed with the induction of 1 mM (final concentration) IPTG at 30 °C for 15 h. After centrifugation, the precipitate of the above bacteria was collected and put into an ultrasonicator on ice to obtain the crude proteins. Finally, rAj-Tspin was purified through affinity chromatography using a nickel column and analyzed by the Tricine SDS-PAGE [44]. Finally, the concentration of rAj-Tspin was measured through Coomassie Brilliant Blue (G250) assay.

### 4.3. Culture of LLC Cells

LLC cells were bought from the National Collection of Authenticated Cell Cultures, thawed at 37 °C and mixed with full high-glucose Dulbecco’s Modified Eagle Medium (DMEM) with 10% fetal bovine serum (FBS, Thermo Fisher Scientific, Waltham, MA, USA) and 1% penicillin-streptomycin at a ratio of 1:1. After centrifugation at 1000 rpm for 1 min and removal of the supernatant, the LLC cells were mixed with the above full-medium and cultured at 37 °C in a CO_2_ incubator for 24 h.

### 4.4. CCK-8 Assay

When the LLC cells grew well, they were digested with trypsin, resuspended with the above full-medium and put into a 96-well plate to culture at 37 °C overnight. After drawing the medium, the LLC cells were cultured in the medium in the presence of 0, 0.37, 0.75, 1.12, 1.49, 1.86, 2.24, 2.98, 3.35 and 3.73 μM rAj-Tspin (final concentration) at 37 °C in the incubator for 24 h. Next, CCK-8 was added into the rAj-Tspin-treated LLC cells and cultured at 37 °C for 4 h. The absorbance of each well at 450 nm (OD value) in the 96-well plate was measured and recorded. Three repetitive CCK-8 assays were performed with empty-medium and rAj-Tspin-treated groups. Finally, the inhibitory rate on LLC cell proliferation was calculated with the following formula: $$\mathrm{The}\;\mathrm{inhibitory}\;\mathrm{rate}\;\mathrm{of}\;\mathrm{proliferation}\;=\frac{{\bar{\mathrm{OD}}}_{\mathrm{Control}\;\mathrm{Groups}}-{\bar{\mathrm{OD}}}_{\mathrm{rAi}-\mathrm{Tspin}-\mathrm{treated}\;\mathrm{Groups}}}{{\bar{\mathrm{OD}}}_{\mathrm{Control}\;\mathrm{Groups}}}\times 100\%$$“—” means average value of OD 

CCK-8 assay was performed as described in the instruction manual of the CCK-8 Kit (Beyotime Biotechnology, Shanghai, China). 

### 4.5. Wright-Giemsa Staining Assay

The digested LLC cells were put onto slides located in the 24-well plate and incubated at 37 °C in the CO_2_ incubator overnight. When the LLC cells grew to 70–80%, different dosages of rAj-Tspin with final concentrations of 1.85, 2.20 and 3.04 μM were injected into the 24-well plate and incubated under the same conditions mentioned previously. Additionally, empty-medium groups were treated as the control groups. After 24 h, the medium in the LLC cells was drawn out completely and solution A in the Wright-Giemsa Stain Kit (Nanjing Jiancheng Bioengineering Institue, Nanjing, China) was added into the 24-well plate immediately and incubated at 37 °C for 5 min in order to fix the LLC cells. Subsequently, solution B in the same kit was also added into the 24-well plate for 10 min. After drawing out the above mixture, the LLC cells on the slides were washed with ultrapure water and observed using a microscope (OLYMPUS, Japan).

### 4.6. TUNEL, Phalloidin-FITC and Hoechst 33258 Staining Assays

First, the LLC cells on the slides in the 24-well plate were pre-treated with the procedures listed in 4.5. Additionally, different dosages of rAj-Tspin (1.85, 2.20, and 3.04 μM) were incubated with the LLC cells at 37 °C for 20 h, with the same volume of empty-medium as the negative controls. Subsequently, both the medium and rAj-Tspin were removed, and the LLC cells were washed with PBS once and fixed with 4% paraformaldehyde for 30 min. When the paraformaldehyde was removed and LLC cells were washed with PBS once, 0.3% TritonX-100 diluted with PBS was used to treat the fixed LLC cells at room temperature for 5 min, and then the above LLC cells were washed with PBS twice in TUNEL assay. TUNEL assay was performed using the TUNEL Apoptosis Assay Kit (Beyotime Biotechnoly, Shanghai, China) according to the manufacture’s protocol. TdT enzyme was mixed with the fluorescent labeling solution at a ratio of 1:9 and then added into the 24-well plates (50 µL/well) at 37 °C for 1.5 h in the dark. After removing the TdT enzyme and the fluorescent labeling solution, the LLC cells on the slides were washed with PBS three times. 

In the cytoskeleton observation assay, the paraformaldehyde-fixed LLC cells were incubated in the dark for 60 min at room temperature with phalloidin-FITC, which was diluted with 1% bovine serum albumin (BSA), to a final concentration of 5 µg/µL. After removing the phalloidin-FITC and washing the LLC cells with PBS three times, Hoechst 33258 was also incubated with the above LLC cells for 5 min. Finally, a laser-scanning confocal microscope (Carl Zeiss, 630×) was used to capture the cytoskeleton and nuclei in the rAj-Tspin-treated LLC cells [45].

### 4.7. Adhesion Assay

The four adhesive molecules, including FN, LN, VN and Col, were diluted with PBS at a final concentration of 0.1 µg/µL and then were respectively coated the 96-well plates (40 µL/well) at 4 °C overnight. The next day, the liquid in the 96-well plate was thrown out, and PBS was used to remove the non-adhesive molecules twice. Then, 1% BSA (100 µL/well) was added into the 96-well plate for blocking. After 1 h, the digested LLC cells were collected and resuspended with empty-medium, then seeded in the 96-well plate (100 µL/well). Furthermore, gradient dosages of rAj-Tspin (0, 1.85, 2.20, and 3.04 μM) were mixed with empty-medium, added into the 96-well plate and incubated with the LLC cells at 37 °C in the CO_2_ incubator. Five hours later, a CCK-8 assay Kit was used to detect the viability of the LLC cells based on the method described in 4.4. Three groups of parallel experiments were performed, and the mean values were calculated [46]. Finally, the adhesion rate of LLC cells was calculated with the formula: $$\mathrm{Adhesive}\;\mathrm{rate}=\frac{{\bar{\mathrm{OD}}}_{\mathrm{rAj}-\mathrm{Tspin}-\mathrm{treated}\;\mathrm{Groups}}\;}{{\bar{\mathrm{OD}}}_{\mathrm{Control}\;\mathrm{Groups}}}\times 100\%$$“—” means average value of OD.

### 4.8. Transwell Assay

Both migration and invasion assays were performed using transwells with similar procedures. The full-medium, which contained bFGF at a final concentration of 3 ng/mL, was pre-incubated at 37 °C, and then 600 µL was added into the lower chamber of each transwell. Next, the upper chamber was located on the plate without inducing bubbles. Then, the resuspended LLC cells were mixed with the empty-medium, which contained 1.85, 2.20 and 3.04 μM rAj-Tspin, respectively, and added into the upper chambers. Additionally, the same volume of empty-medium was used to treat the LLC cells, which was taken as the control group. After 20 h, the upper chambers were taken out carefully, and the liquid was drawn out. When the non-migrated LLC cells were wiped off, the migrated LLC cells on the polycarbonate membranes were fixed for 10 min. Subsequently, the polycarbonate membranes were put onto the glass slides and stained with Wright-Giemsa dye, respectively. Finally, cover slips were placed on the top of the membranes, and three fields were randomly selected and counted. Before adding the LLC cells, 40 µL Matrigel solution at a final concentration of 4 mg/mL was added into the upper chamber of the transwells and incubated at 37 °C for 30 min to allow the Matrigel to curdle in the invasion assay. The invasive experiments were incubated at 37 °C for 40 h with other steps, which were similar to those followed in the migration assay [47].

### 4.9. Statistics

Results are expressed as mean ± SE. Statistical significance of differences in means between groups were determined using analysis of variance. All experiments were repeated at least three times. Statistical significance was shown as follows: * *p* < 0.05; ** *p* < 0.01; and *** *p* < 0.001.

## 5. Conclusions

In conclusion, rAj-Tspin is a recombinant peptide cloned from the third TSP1 domain of ADAMTS13-like from sea cucumber (*A. japonicus*). In addition to the characteristics of the TSP1 domain, rAj-Tspin also has an RGD motif. Although Aj-Tspin shares low homology with disintegrins, soluble rAj-Tspin showed anti-tumor effects on LLC cells, including inhibitory roles on proliferation, adhesion, migration and invasion, induction of apoptosis and destruction of the cytoskeleton. All above results indicate that rAj-Tspin might act as a potential anti-tumor drug during the treatment of NSCLC patients in the near future.

## 6. Patents

Recombinant peptide from sea cucumber and its application in the preparation of antitumor drugs, patent number ZL 2019 1 1182029. 5.

## Figures and Tables

**Figure 1 molecules-27-00229-f001:**
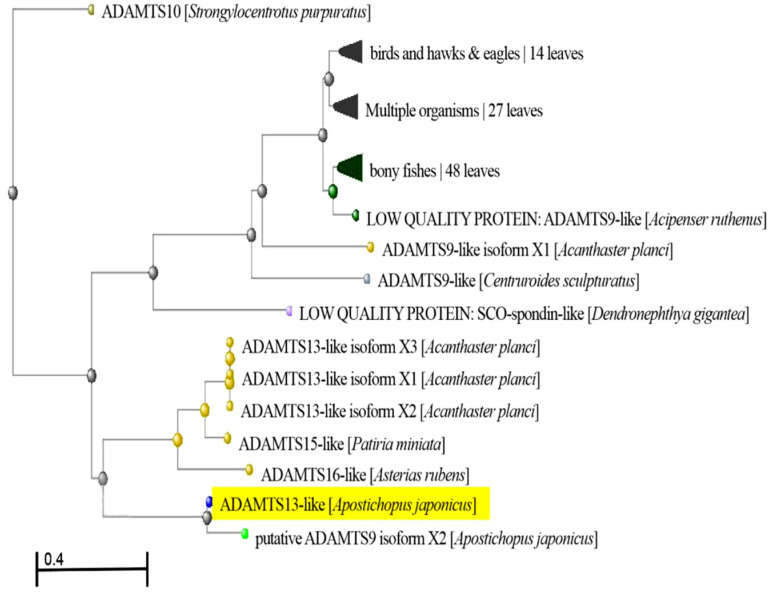
Phylogenetic tree analysis of ADAMTS13-like from *A. japonicus*.

**Figure 2 molecules-27-00229-f002:**
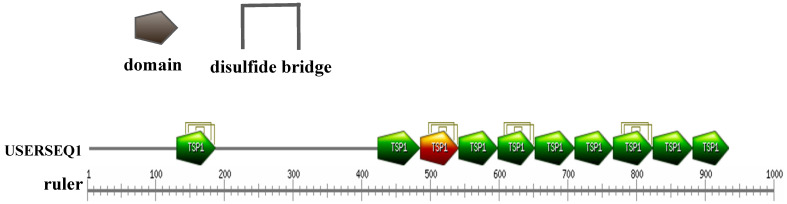
Domain analysis of ADAMTS13-like from *A. japonicus*.

**Figure 3 molecules-27-00229-f003:**
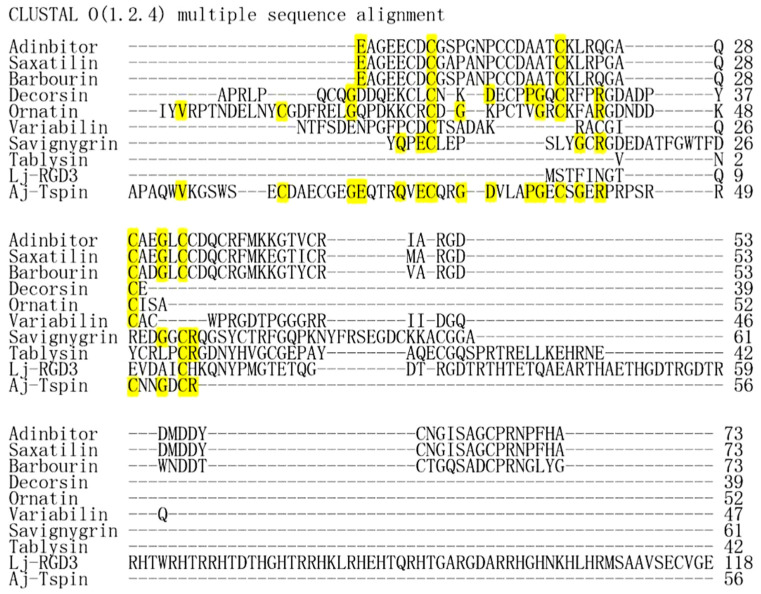
Multiple-sequence alignment of disintegrins from snakes, leeches, ticks, gadflies and lampreys, as well as Aj-Tspin from sea cucumbers. Adinbitor, saxatilin, barbourin, decorsin, ornatin, variabilin, savignygrin, tablysin, Lj-RGD3 and Aj-Tspin are, respectively, from *Agkistrodon halys brevicaudus stejneger*, *Gloydius saxatilis*, *Sistrurus miliarius barbouri*, *Macrobdella decora*, *Placobdella ornata*, *Dermacentor variabilis*, *Ornithodoros savignyi*, *Tabanus yao*, *Lampetra japonica* and *A. japonicus*.

**Figure 4 molecules-27-00229-f004:**
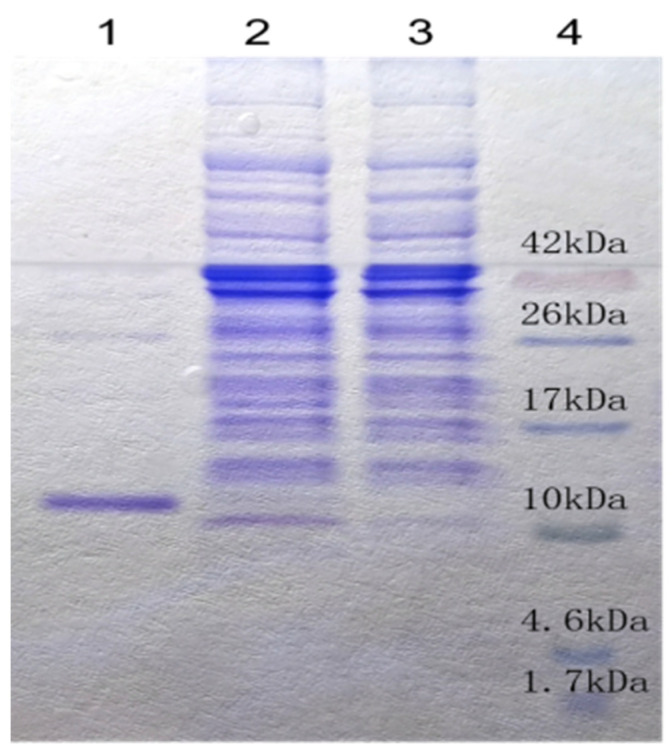
Purified rAj-Tspin was detected on Tricine SDS-PAGE. Lane 1: 10 μL of purified rAj-Tspin; Lane 2: IPTG-induced expression of recombinant BL21 cells; Lane 3: non-induced expression of recombinant BL21 cells; Lane 4: spectra multicolor low range protein ladder (Thermo Scientific, Waltham MA, USA).

**Figure 5 molecules-27-00229-f005:**
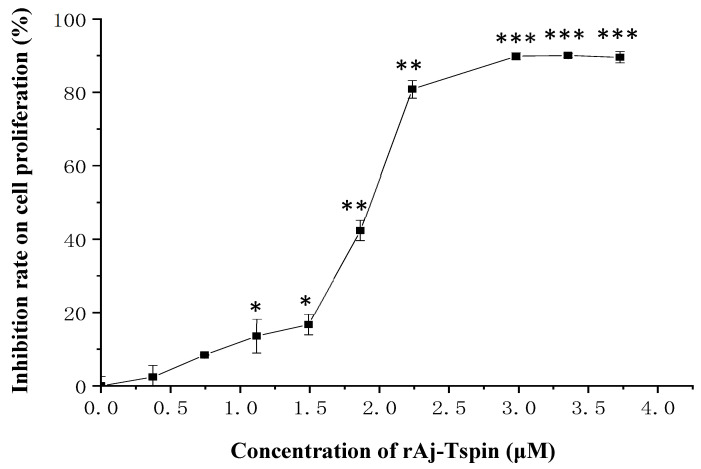
CCK-8 assay detected the inhibitory effects of rAj-Tspin on the proliferation of LLC cells. *, ** and ***, respectively, indicate *p* < 0.05, *p* < 0.01, and *p* < 0.001 (compared with control group). Each assay was repeated three times.

**Figure 6 molecules-27-00229-f006:**
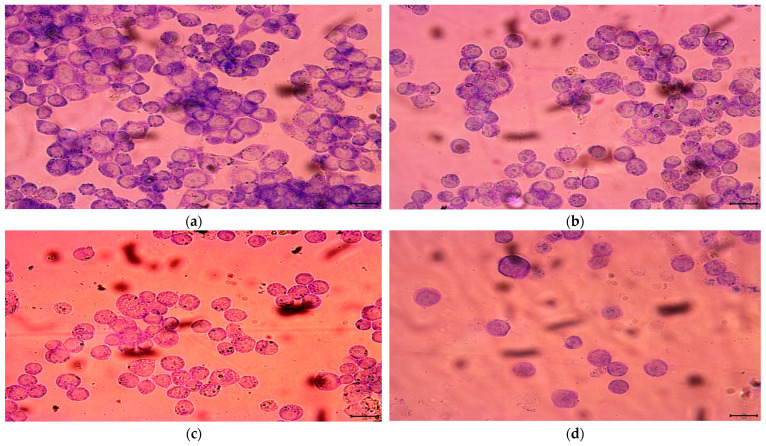
Wright-Giemsa staining assay detected the effect of rAj-Tspin on the morphology of LLC cells. The bar indicates 10 μm; (**a**) control; (**b**) 1.85 μM rAj-Tspin; (**c**) 2.20 μM rAj-Tspin; (**d**) 3.04 μM rAj-Tspin.

**Figure 7 molecules-27-00229-f007:**
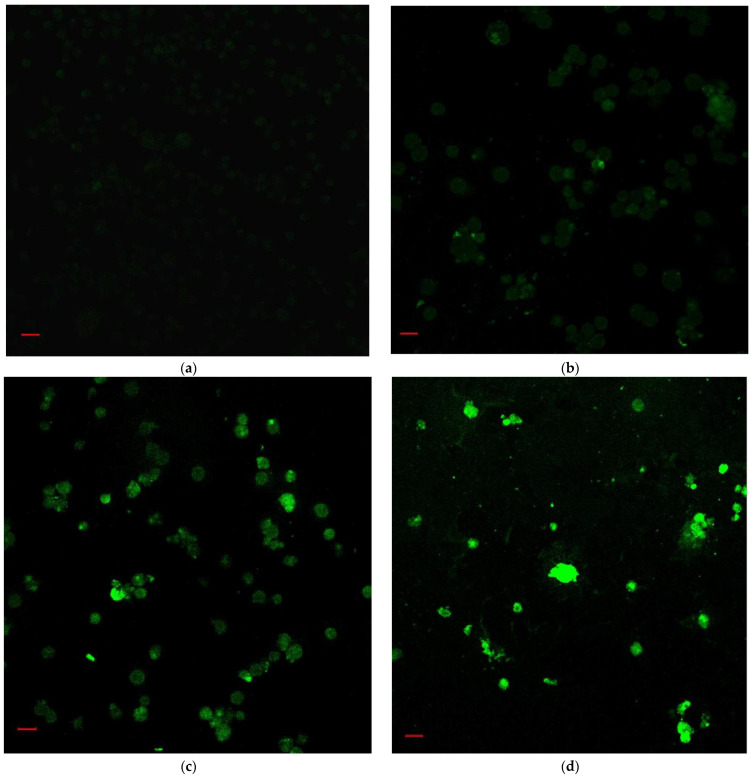
The apoptosis of LLC cells induced by rAj-Tspin was detected by TUNEL assay (Zeiss laser-scanning confocal microscopy, 200×). The bar indicates 20 μm; (**a**) control; (**b**) 1.85 μM rAj-Tspin; (**c**) 2.20 μM rAj-Tspin; (**d**) 3.04 μM rAj-Tspin.

**Figure 8 molecules-27-00229-f008:**
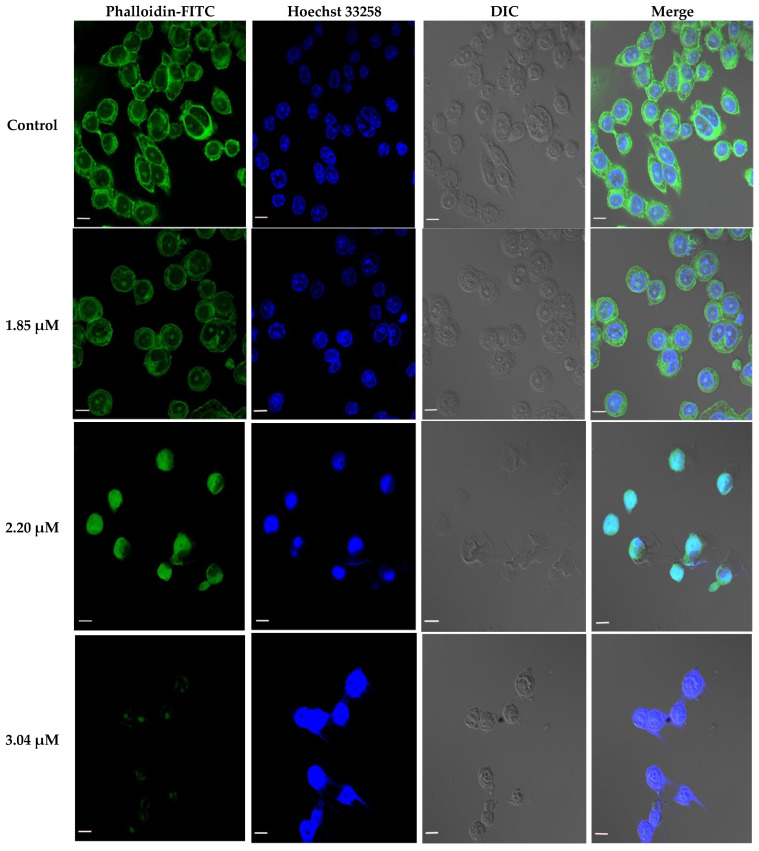
Effect of rAj-Tspin on the cytoskeleton and nuclei of the LLC cells. Green indicates the cytoskeleton marked by the phalloidin-FITC. Blue indicates the nuclei marked by Hoechst 33258 (Zeiss laser-scanning confocal microscopy, 630×). The bar indicates 10 μm.

**Figure 9 molecules-27-00229-f009:**
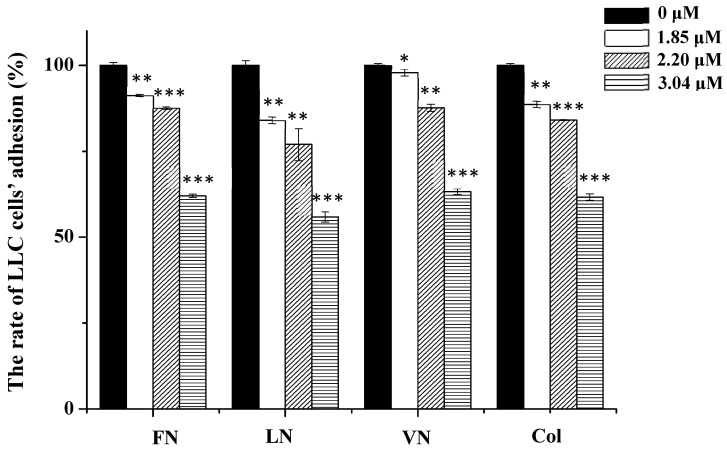
rAj-Tspin inhibited the adhesion of LLC cells. *, ** and ***, respectively, indicate *p* < 0.05, *p* < 0.01, and *p* < 0.001 (compared with the empty-medium groups).

**Figure 10 molecules-27-00229-f010:**
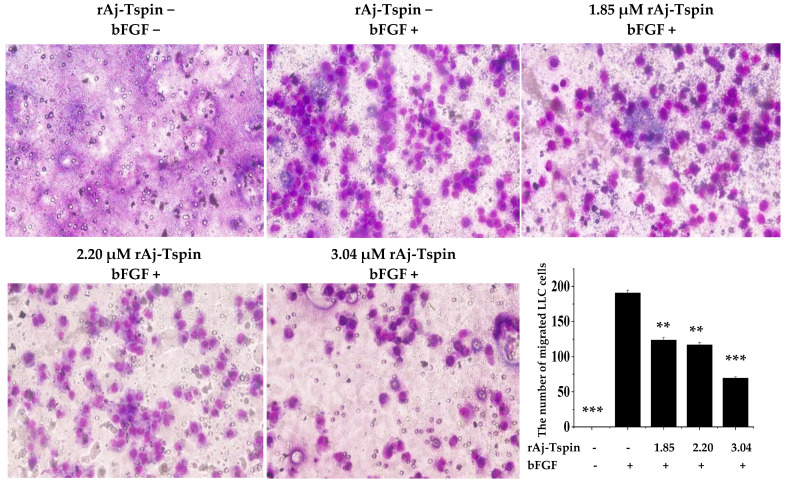
rAj-Tspin inhibited the migration of LLC cells (OLYMPUS optical microscopy, 200×). ** and ***, respectively, indicate *p* < 0.01 and *p* < 0.001 (compared with the empty-medium and bFGF treatment groups).

**Figure 11 molecules-27-00229-f011:**
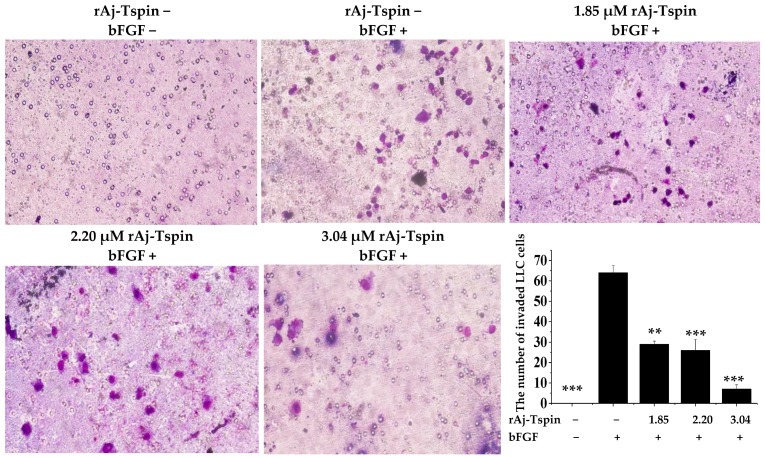
rAj-Tspin inhibited the invasion of LLC cells (OLYMPUS optical microscopy, 200×). ** and ***, respectively, indicate *p* < 0.01 and *p* < 0.001 (compared with the empty-medium and bFGF treatment groups).

**Table 1 molecules-27-00229-t001:** Adhesive rates of LLC cells to the four ECM proteins in the presence of rAj-Tspin.

rAj-Tspin (μM).	0	1.85	2.20	3.04
ECM Proteins	Adhesive Rates			
FN	100.00	91.24	87.61	62.03
LN	100.00	84.02	76.99	55.88
VN	100.00	97.89	87.62	63.24
Col	100.00	88.66	84.06	61.67

## Data Availability

Not applicable.

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
