# Peer review of "Cloning, Expression and Inhibitory Effects on Lewis Lung Carcinoma Cells of rAj-Tspin from Sea Cucumber (Apostichopus japonicus)"

_molecules, 2021, doi:10.3390/molecules27010229_

Round 1
Reviewer 1 Report
This paper presents the role of rAj-Tspin from sea cucumber on lung carcinoma cells. The manuscript is well written in general, but it needs some improvements, not really linked to the scientific part, but the presentation of this study.
The title should not include abbreviated words. The abstract should include some info on the background and the importance of this study. The English should be verified and the editing (see lines 36-38, 43-44, 75-77, 270). The authors must verify the references (no 1, for example, has no details).
Reviewer 2 Report
The manuscript titled “Cloning, Expression and Inhibitory Effects on LLC Cells of rAj-Tspin from Sea Cucumber (Apostichopus japonicus)” by Rong et al.,
Although the manuscript state clearly why was it undertaken, but clinical application was not clearly stated or discussed.
Manuscript is poorly presented with several grammatical as well as typological error and hence, it is recommended to go through the manuscript clearly to rectify these issues. Some of the points are mentioned below:
Line 15: Apostichopus japonicus change it to Italics
Abstract is not clear: please rewrite it in more concise way. For example, “In order to screen the anti-tumoral proteins or peptides from sea cucumber (Apostichopus 15 japonicus), its cDNA library was analyzed and a disintegrin-like and metalloproteinase with thrombospondin type 1 motif, member 13 (ADAMTS13)-like which contains 10 thrombospondin 1 (TSP1) domains was found.” – State clearly and divide the sentence
Line 60: Apostichopus japonicus, A. japonicus- Change to italics
“In recent years, sea cucumber has become a favorite health care food due to its characterization on preventing cardiovascular diseases, suppressing tumors, as well as enhancing immunities.”- Provide references.
Line 81, 83: A. japonicus- change to Italics (Check throughout the manuscript, also other names should be italicised).
Figure 1 and 4. Should be transferred to SI
Was any of the sequence submitted to public repository? If so provide the accession number.
How was the statistical analysis performed to evaluate the data?
None of the methodology for in vitro experiments provides citations. Also mention in details about negative and positive controls used in the experiment.
Conclusion should be provided in more elaborate form.
Round 2
Reviewer 1 Report
The manuscript was improved in many aspects, but there could be some other small adjustments needed.
The firsts 2 sentences may be revised as "disintegrin-like and metalloproteinase with thrombospondin motif" is repeated twice.
I recommend inserting a paragraph at the end of the discussion with the strengths and limits of this study, and more importantly, with the significance of this study from a clinical point of view or further possibilities to continue it.
Author Response
- The firsts 2 sentences may be revised as "disintegrin-like and metalloproteinase with thrombospondin motif" is repeated twice.
Re: Thank you for your suggestion. And the English has been verified and revised. ( Page 1, line 41-42, highlight in green).
- I recommend inserting a paragraph at the end of the discussion with the strengths and limits of this study, and more importantly, with the significance of this study from a clinical point of view or further possibilities to continue it.
Re: Thank you for your suggestion. And we have inserted a paragraph at the end of discussion with the strengths and limits of this study, and the significance of this study and further possibilities (Page 12, line 293-300, highlight in green).

Reviewer 2 Report
The authors have provided the response to all the questions raised and also the manuscript has been improved substantially. Hence, I recommend it for publication in the current form.
Author Response
Questions from the Reviewer 2
The authors have provided the response to all the questions raised and also the manuscript has been improved substantially. Hence, I recommend it for publication in the current form.
Re: Thank you for your suggestion and help. Thank you very much for your final approval of our revised manuscript.